# Spontaneous Occurrence of Various Types of Hepatocellular Adenoma in the Livers of Metabolic Syndrome-Associated Steatohepatitis Model TSOD Mice

**DOI:** 10.3390/ijms231911923

**Published:** 2022-10-07

**Authors:** Wenhua Shao, Orgil Jargalsaikhan, Mayuko Ichimura-Shimizu, Qinyi Cai, Hirohisa Ogawa, Yuko Miyakami, Kengo Atsumi, Mitsuru Tomita, Mitsuko Sutoh, Shunji Toyohara, Ryoji Hokao, Yasusei Kudo, Takeshi Oya, Koichi Tsuneyama

**Affiliations:** 1Department of Molecular Pathology, Institute of Biomedical Sciences, Tokushima University Graduate School, Tokushima 770-8503, Japan; 2Department of Oral Bioscience, Tokushima University Graduate School of Biomedical Sciences, Tokushima 770-8503, Japan; 3Department of Pathology and Laboratory Medicine, Institute of Biomedical Sciences, Tokushima University Graduate School, Tokushima 770-8503, Japan; 4Institute for Animal Reproduction, Ibaraki 300-0131, Japan

**Keywords:** glutamine synthetase, liver fatty acid-binding protein, serum amyloid A, beta-catenin, malignant transformation, focal nodular hyperplasia, immunostaining, diabetes, hyperlipidemia, HCA-like tumor

## Abstract

Male Tsumura-Suzuki Obese Diabetes (TSOD) mice, a spontaneous metabolic syndrome model, develop non-alcoholic steatohepatitis and liver tumors by feeding on a standard mouse diet. Nearly 70% of liver tumors express glutamine synthetase (GS), a marker of hepatocellular carcinoma. In contrast, approximately 30% are GS-negative without prominent nuclear or structural atypia. In this study, we examined the characteristics of the GS-negative tumors of TSOD mice. Twenty male TSOD mice were sacrificed at 40 weeks and a total of 21 tumors were analyzed by HE staining and immunostaining of GS, liver fatty acid-binding protein (L-FABP), serum amyloid A (SAA), and beta-catenin. With immunostaining for GS, six (29%) tumors were negative. Based on the histological and immunohistological characteristics, six GS-negative tumors were classified into several subtypes of human hepatocellular adenoma (HCA). One large tumor showed generally similar findings to inflammatory HCA, but contained small atypical foci with GS staining and partial nuclear beta-catenin expression suggesting malignant transformation. GS-negative tumors of TSOD mice contained features similar to various subtypes of HCA. Different HCA subtypes occurring in the same liver have been reported in humans; however, the diversity of patient backgrounds limits the ability to conduct a detailed, multifaceted analysis. TSOD mice may share similar mechanisms of HCA development as in humans. It is timely to review the pathogenesis of HCA from both genetic and environmental perspectives, and it is expected that TSOD mice will make further contributions in this regard.

## 1. Introduction

We previously reported that male Tsumura-Suzuki Obese Diabetes (TSOD) mice, a spontaneous metabolic syndrome model, developed non-alcoholic steatohepatitis and liver tumors at a high rate after 8 months of age by feeding on a standard mouse diet [1,2,3]. TSOD mice were inbred strains created from ddY mice by mating with obese and diabetic individuals for more than 30 generations. Meanwhile, inbred Tsumura-Suzuki non-Obese (TSNO) mice were inbred strains produced from ddY mice by mating with non-obese, non-diabetic individuals for more than 30 generations. TSNO mice were used as a normal control group in this experiment. We previously found that nearly 70% of liver tumors diffusely expressed glutamine synthetase (GS), a marker of hepatocellular carcinoma (HCC), regardless of size, and that these GS-positive tumors had characteristics of human dysplastic nodules of HCC. In contrast, about 30% of the liver tumors appearing in TSOD mice were GS-negative, most of which had weak nuclear and structural atypia, resembling human hepatocellular adenoma (HCA) [1].

Hepatocellular adenoma is a rare benign tumor of the non-cirrhotic liver that has been associated with oral contraceptive use in young women, steroid use, and type I and type III glycogenic disease [4]. According to reports from Europe and the United States, its incidence is 3–4 per 100,000 population, with 85% of cases occurring in women of childbearing age. It is rare in children, men, and those aged >65 years. In recent years, cases of HCA have also arisen due to obesity and metabolic syndrome, and it has been predicted that the number of such patients may increase in the future [5,6]. The diagnosis of HCA requires differentiation from focal nodular hyperplasia (FNH), which is a non-neoplastic lesion, as well as from HCC, which is often difficult to diagnose based on the histologic findings alone. However, it has become clear that the classification of subtypes based on genetic abnormalities is associated with phenotypes using immunostaining, which has greatly improved the accuracy of HCA diagnosis. In addition, it is now known that the risk of malignant transformation differs among subtypes of HCA, increasing the need for accurate subtyping.

Most cases of HCA show a benign course, but hemorrhage and malignant transformation have been reported in rare cases [7,8,9,10]. The pathogenesis of HCA remains unclear. The development of animal models useful for elucidating the pathogenesis is anticipated, but, with the exception of glycogenic disease models, there are few useful models yet available for such analysis [11,12,13,14,15].

According to the 2019 WHO classification, HCA is divided into four categories: HNF1α-inactivated HCA (H-HCA), inflammatory HCA (IHCA), beta-catenin-activated HCA (b-HCA), and β-catenin-activated IHCA (b-IHCA) [4,16]. Other subtypes, such as sonic hedgehog HCA (shHCA) and ASS1-positive HCA due to overexpression of the ASS1 enzyme, have also been reported [17]. Of these, H-HCAs and IHCAs comprise the majority, with H-HCAs accounting for 30–35% of all HCAs.

### 1.1. HNF1α-Inactivated HCA (H-HCA)

Mutations in the HNFIA gene, which encodes HNF1α, a transcription factor involved in hepatocyte differentiation, result in the loss of protein expression in tumor cells. In immunostaining, antibodies against the liver fatty acid-binding protein (L-FABP), which is positively regulated by the HNF1A gene, are useful diagnostic markers. In this type of HCA, L-FABP is attenuated or shows as negative [18]. The tumor shows a lobular outline, typically with diffuse large or small fatty droplets, a ballooning and clear cell appearance, and, rarely, pseudoglandular structures [19].

### 1.2. Inflammatory HCA (IHCA)

Inflammatory HCAs account for 35–40% of all HCAs and are the most common subtype [20]. Mutations that activate the IL-6/JAK/STAT3 pathway may result in IHCA development. The expression of acute-phase inflammation-related proteins such as serum amyloid A (SAA) and C-reactive protein (CRP) is increased at the mRNA and protein levels. IHCA typically presents with dilated sinusoids, with congestion, inflammation, and clusters of wall-thickened small arteries surrounded by extracellular matrix, as well as pseudoportal areas with a bile ductular reaction. Partial fatty degeneration is often observed. The histologic features of IHCA are similar to those of FNH, and distinction between the two is often problematic. Although it is believed that patients with IHCA without β-catenin mutations do not develop malignant transformation, a case of malignant transformation has been reported in a patient with glycogenic disease without β-catenin mutation [8].

### 1.3. β-Catenin-Activated HCA (b-HCA)

Mutations or deletions in the CTNNB1 (cadherin-associated protein B1) gene encoding β-catenin have been reported to cause activated WNT signaling and tumorigenesis in b-HCA. It is reported that b-HCAs account for 10% of all HCAs and that b-IHCAs, described below, account for 10–15% [5,21]. Immunostaining for β-catenin antibodies shows positive findings at the cell membrane of both tumor and non-tumor cells, but this subtype shows ectopic expression in the nuclei of tumor cells [18].

### 1.4. β-Catenin-Activated IHCA (b-IHCA)

b-IHCA is considered to have features of both IHCA and b-HCA and is at risk for malignant transformation [4,22].

In this study, we examined in detail the characteristics of GS-negative tumors that spontaneously develop in TSOD mice against the background of metabolic syndrome. We aimed to clarify the dissimilarities between these tumors and human hepatocellular adenomas based on histopathological and immunohistological analyses. The purpose of this study is to propose the utility of the TSOD mouse as an animal model that can help elucidate the natural history of HCA and its pathogenesis, which is still largely unknown.

## 2. Results

### 2.1. Frequency of Liver Tumor

Grossly, there were no liver tumors in TSNO mice, while 19/20 (95%) of TSOD mice examined had one or more liver tumors. One section containing the largest tumor in each of the 19 grossly positive individuals and one largest section of the tumor-free individual were prepared for histopathological analyses. The 20 specimens from 20 individuals were observed microscopically, and all tumors of size ≥1 mm were evaluated. One to two tumors were identified in each section, and a total of 21 tumors were analyzed in detail. In immunostaining for GS, 15 (71%) tumors were GS-positive and six (29%) tumors were GS-negative.

### 2.2. Characteristic Histopathological and Immunohistochemical Findings of GS-Positive Liver Tumors

The histopathological characters of the GS-positive tumors were as previously reported [1,23,24]. Tumors comprised atypical hepatocytes in an irregular trabecular structure. Fibrous capsules were not observed. A substitutional growth pattern was observed in the border to background liver and portal invasion was frequently observed (Figure 1A–C). Immunohistochemically, no GS-positive tumor showed L-FABP attenuation (Figure 1D,E). Taken together, the GS-positive tumors were considered to include lesions in the sequence of progression from dysplastic nodule to well-differentiated HCC.

### 2.3. Characteristics of GS-Negative Liver Tumors

GS-negative tumors showed various histological findings mimicking human hepatocellular adenoma. Based on the histological and immunohistological characteristics, the six tumors examined in this study were classified into four types similar to those of human hepatocellular adenoma. The characteristics of each type are described below and listed in Table 1.

#### 2.3.1. HNF1a-Inactivated HCA Tumor Mimics

Two tumors (tumors 3 and 5) showed similar pathological and immunohistochemical findings to human HNF1 a-inactivated HCA. Hepatocytes showed trabecular structure, and individual cellular atypia was unremarkable. Some tumor cells contained small lipid droplets in the cytoplasm. An invasive growth pattern was not observed (Figure 2 A,B,G,H). Immunohistochemically, these tumors showed reduced expression of L-FABP. Intense SAA expression was not observed. β-catenin was expressed only in the cell membrane (Figure 2C–F,I–L).

#### 2.3.2. Inflammatory HCA Tumor Mimics

Two tumors (tumors 2 and 4) showed similar pathological and immunohistochemical findings to human inflammatory HCA. Hepatocytes showed trabecular structure with dilatation of sinusoids. Pseudoglandular hepatocellular arrangements were occasionally observed, in addition to thick-walled vessels and sparse aggregation of inflammatory cells. An invasive growth pattern was not observed (Figure 3A,B,G,H). Immunohistochemically, these tumors did not show reduced expression of L-FABP. Some intense SAA expression was observed in one case. Another case did not show intense SAA expression; however, the histopathological characteristics were quite similar to those of inflammatory HCA. β-catenin was expressed only in the cell membrane (Figure 3C–F,I–L).

#### 2.3.3. Both HNF1 α-Inactivated HCA and Inflammatory HCA Tumor Mimic

One tumor (tumor 1) shared the pathological and immunohistochemical findings of human HNF-1 alpha-inactivated HCA and inflammatory HCA. Pathologically, hepatocytes showed trabecular structure, and individual cellular atypia was unremarkable. There were no clear lipid droplets in the cytoplasm. Sinusoidal dilatation and inflammatory cell aggregation was not observed. An invasive growth pattern was not seen (Figure 4A,B). Immunohistochemically, these tumors showed reduced expression of L-FABP with intense SAA expression. b-catenin was expressed only in the cell membrane (Figure 4C–F).

#### 2.3.4. B-Catenin-Activated IHCA (B-IHCA) Tumor Mimic

One tumor (tumor 6) showed generally similar pathological and immunohistochemical findings to human inflammatory HCA (Figure 5A–D). However, small atypical foci (2 mm in size) showing nuclear atypia of hepatocytes with a partial thick trabecular pattern are included as characteristics of inflammatory HCA (Figure 5B,D). Immunohistochemically, most of the tumor did not show GS immunostaining or reduced expression of L-FABP. Intense SAA expression was observed, and b-catenin was expressed only in the cell membrane (Figure 5E–H). In contrast, small atypical foci showed intense and diffuse GS staining. Reduced expression of L-FABP and intense SAA expression was not observed (Figure 5I–K). As noted, b-catenin was expressed in the nucleus of almost half of the tumor cells with small atypical foci (Figure 5L). Although these atypical foci did not show an invasive growth pattern, the histopathological findings indicated that this tumor was in the process of malignant transformation.

## 3. Materials and Methods

Twenty male TSOD mice and TSNO mice were fed on a standard mouse diet and sacrificed at 40 weeks of age. The livers were immediately removed and soaked in 10% neutral buffered formalin, cut surgically into 3-mm-wide whole sections, and the tumors were confirmed grossly. From each individual, one section containing the largest tumor was prepared and subjected to HE staining and immunohistochemical analysis of GS, L-FABP, SAA, and β-catenin. Tumors of size ≤ 1 mm microscopically were included for evaluation.

All institutional and national guidelines for the care and use of laboratory animals were followed. Furthermore, this study was performed following the animal experiment guidelines specified in the Institute for Animal Reproduction (Ibaraki, JAPAN), which strictly abides by the rules of guidance on animal research ethics from the International Association of Veterinary Editors’ Consensus Author Guidelines on Animal Ethics and Welfare. Approval code from the Experimental Animal Welfare Ethics Committee of Institute for Animal Reproduction is IAR-EAWEC-2021-2-61, which was approved on 18 January 2021. This article does not involve any studies with human subjects.

## 4. Conclusions

This study revealed that TSOD mice spontaneously develop tumors similar to human HCA, in addition to HCC. HCA is defined as occurring in the normal liver, but a number of tumors with features similar to those of HCA have been reported in the context of alcoholic liver injury and other conditions, and are referred to by various names, including “HCA-like tumor”. The difference between HCA and HCA-like tumor is still inconclusive, as they are considered by some as different disease entities, and by others as having the same characteristics. The background liver of the TSOD mice studied in this study showed various degrees of NASH. In this sense, the term “HCA-like tumor” may be accurate. However, the biological and histopathological characteristics of HCA-like tumors are still unknown, which is why we compared liver tumors of TSOD mice with various subtypes of HCAs in humans.

Interestingly, we observed the occurrence of liver tumors with various morphological and immunohistological features, including tumors that are similar to human HNF1 a -inactivated HCA, inflammatory HCA, and b-catenin-activated IHCA (b-IHCA). In humans, HCAs often appear as multiple tumors, and reports of different HCA subtypes occurring in the same liver have been observed. However, few human cases have been reported, and the diversity of patient backgrounds limits the ability to conduct a detailed, multifaceted analyses. TSOD mice are a spontaneous metabolic syndrome model with a background of obesity, hyperlipidemia, and diabetes mellitus under normal feeding conditions and without the need for special treatment, suggesting shared mechanisms with the development of HCA in humans.

There are several reports of mouse models that developed HCA. For example, HCA developed in a mouse model of glycogen storage disease [14,25] and liver tumors similar to inflammatory HCA developed in NF-κB/RelA double knockout mice [26]. Amano et al. reported a model in which both HCC and HCA occurred simultaneously in a NASH background by feeding a special diet to a genetically engineered model [27]. In this sense, the mouse model showing HCA is not novel. However, TSOD mice have the advantage of spontaneously developing a variety of HCAs in addition to HCC without the need for genetic modification or special diets. The following future examination should be expected to take advantage of this feature.

### 4.1. Investigation of the Molecular Biological Features and Microenvironmental Differences Underlying the Development of HCA Subtypes

The present analysis of 20 TSOD mice identified at least three subtypes of HCA sharing characteristics of multiple subtypes. Although the molecular basis for each HCA subtype in humans is well established and characteristic histological findings have been accumulated, the events that trigger the development of these tumors remain unclear. Unique to the TSOD mouse model, each subtype is represented by multifaced characteristics, including molecular biological information and immunological aspects with surrounding hepatocytes, that can be compared with each other. Formalin fixation was performed for all liver tumors in the present study. Simultaneous preparation of frozen specimens would enable additional molecular biological analysis and immune cell kinetics analysis.

### 4.2. Investigation of Carcinogenic Mechanism for HCA

Among the liver tumors identified in this study, one tumor (tumor 6) contained a localized atypical focus within inflammatory HCA. This tumor may be a very useful subject for analysis of the malignant transformation of HCA. We have recently developed a new liver biopsy technique that enables liver tissue to be obtained from the same individual up to four times over time. Tumor 6 was a large tumor, sized approximately 3–15 mm. Targeted liver biopsy of a large liver tumor such as this over time may allow the collection of histological samples of HCA that have a high probability of malignant transformation over time. Analysis of the properties of such tumors over time at the molecular level would be expected to help elucidate the mechanism of the malignant transformation of HCA.

As the pathogenesis of HCA becomes clearer, reports of unusual cases such as multiple occurrence or non-normal background liver are increasing. At this point it is timely to review the pathogenesis of HCA from both genetic and environmental perspectives, and it is expected the TSOD mice will make further contributions in such investigations.

In conclusion, TSOD mice are expected to be used to analyze how the metabolic syndrome is involved in the development of HCA. In addition, long-term observation will help to elucidate the mechanism of the malignant transformation of HCA.

## Figures and Tables

**Figure 1 ijms-23-11923-f001:**
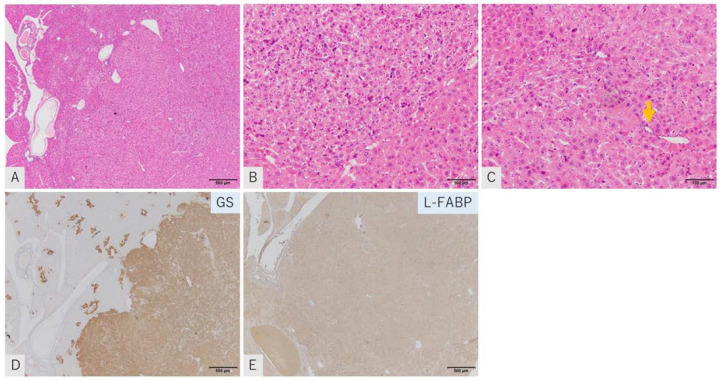
Representative images of GS-positive tumors. HE staining (**A**–**C**), immunostaining of GS (**D**) and L-FABP (**E**). Tumor shows a thick trabecular pattern of atypical hepatocytes with no capsule. Remnant bile ducts due to portal invasion are seen (**C**: arrow). Tissue is positive for GS (**D**) and for L-FABP (**E**) (scale: **A**,**D**,**E**: 500 μm; **B**,**C**: 100 μm).

**Figure 2 ijms-23-11923-f002:**
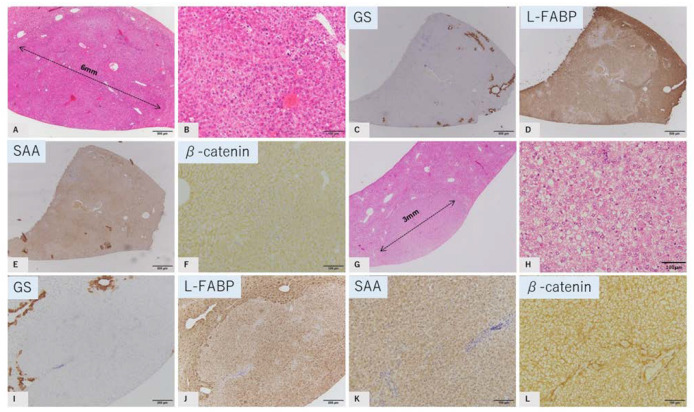
GS-negative tumor mimics of HNF1 a-inactivated HCA. Tumor 3, (**A**–**F**) and Tumor 5, (**G**–**L**). HE staining (**A**,**B**,**G**,**H**); immunostaining of GS (**C**,**I**), L-FABP (**D**,**J**), SAA (**E**,**K**), and b-catenin (**F**,**L**). Tumor shows a trabecular pattern of hepatocytes showing mild vacuolation. No capsule is observed. GS: negative, L-FABP: attenuated, SAA: no strong staining, b-catenin: membranous staining. Tumor diameter is 6 mm in tumor 3 (**A**), and 3 mm in tumor 5 (**G**) (scale: **C**–**F**,**I**–**L**, 500 μm; **B**,**H**, 100 μm).

**Figure 3 ijms-23-11923-f003:**
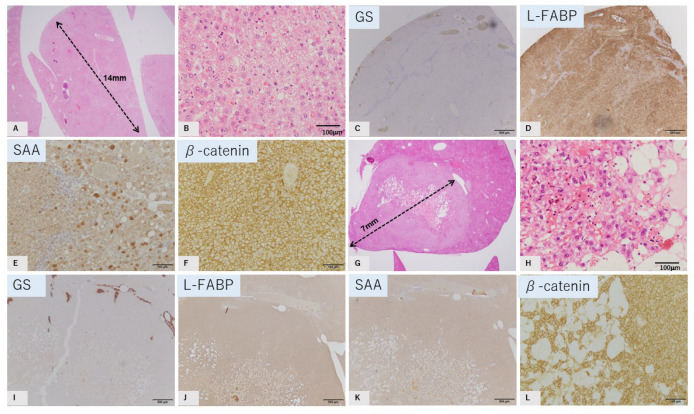
GS-negative tumor mimics of inflammatory HCA. Tumor 2, (**A**–**F**); and (**G**–**L**): tumor 4. HE staining (**A**,**B**,**G**,**H**). Immunostaining of GS (**C**,**I**), L-FABP (**D**,**J**), SAA (**E**,**K**), and b-catenin (**F**,**L**). Tumor shows a trabecular pattern of hepatocytes with mild vacuolation. Occasional sinusoidal dilatation was observed with sparse aggregation of inflammatory cells. No capsule is observed. GS: negative, L-FABP: positive, SAA: strong positive staining is observed in parts of (**E**) and no strong staining was seen in (**K**), b-catenin: membranous staining. Tumor diameter is 14 mm in tumor 2 (**A**) and 7 mm in tumor 4 (**G**) (scale: **C**–**F**,**I**–**L**, 500 μm; **B**,**H**, 100 μm).

**Figure 4 ijms-23-11923-f004:**
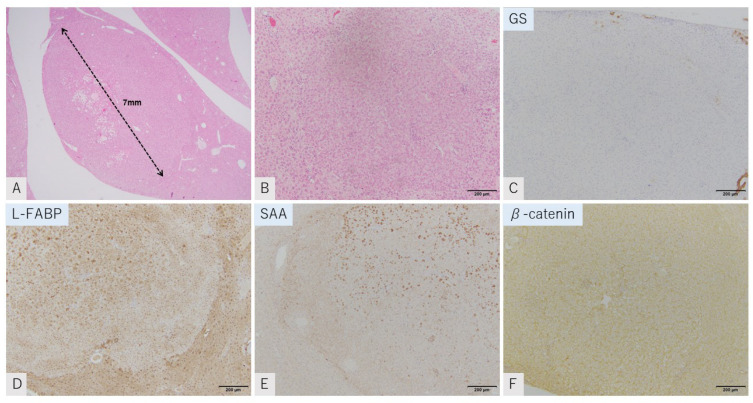
GS-negative tumor mimic of both HNF1 a-inactivated HCA and inflammatory HCA (tumor 1). HE staining (**A**,**B**). Immunostaining of GS (**C**), L-FABP (**D**), SAA (**E**), and b-catenin (**F**). Tumor shows a trabecular pattern of hepatocytes with mild vacuolation. Sinusoidal dilatation is observed. There is sparse aggregation of inflammatory cells. No capsule is observed. GS: negative, L-FABP: attenuated, SAA: strong positive staining was observed in parts, b-catenin: membranous staining. Tumor diameter is 7 mm in (**A**) (scale: **B**–**F**, 20 μm).

**Figure 5 ijms-23-11923-f005:**
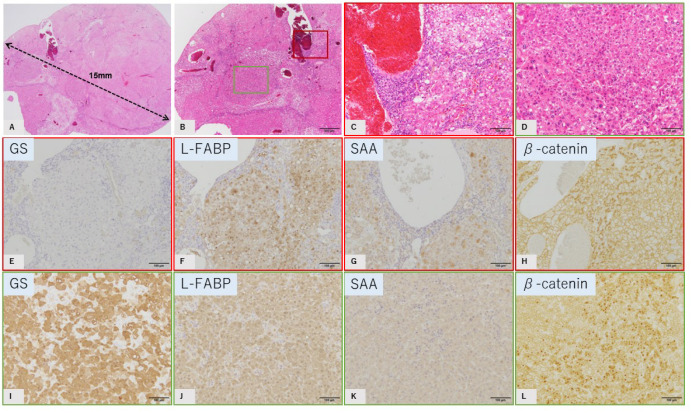
GS-negative tumor mimics of b-catenin-activated IHCA (b-IHCA) (tumor 6). HE staining (**A**–**D**). Immunostaining of lesions seen in C (**E**–**H**) and D (**I**–**L**), which are indicated by the colored boxes in (**B**). Immunostaining of GS (**E**,**I**), L-FABP (**F**,**J**), SAA (**G**,**K**), and b-catenin (**H**,**L**). (**A**,**B**): Tumor diameter is 15 mm and various histological patterns are seen. Most of the tumor has a trabecular pattern of hepatocytes with minimal atypia and sinusoidal dilatation, with some small atypical foci of diameter 2 mm. (**C**): Representative image of the main part of the tumor (higher magnification of the red box in (**B**)). (**D**): Representative image of atypical foci (higher magnification of the green box in (**B**)). Immunostaining of GS: negative in (**E**) and diffuse positive in (**I**); L-FABP: positive in (**F**,**J**); SAA: no strong staining in (**G**,**K**); b-catenin: membranous staining in (**H**), nuclear staining in (**L**). Tumor diameter is 15 mm (**A**) (scale: **B**, 500 μm; **C**–**L**, 100 μm).

**Table 1 ijms-23-11923-t001:** Characteristics of GS-negative tumors.

No	Diameter	GS	L-FABP	SAA	β-Catenin
1	7 mm	Negative	Reduced	Positive	Membrane
2	14 mm	Negative	Positive	Positive	Membrane
3	6 mm	Negative	Reduced	Negative	Membrane
4	7 mm	Negative	Positive	Negative	Membrane
5	3 mm	Negative	Reduced	Negative	Membrane
6: Major	15 mm	Negative	Positive	Positive	Membrane
6: focal	2 mm × 2	Positive	Positive	Negative	Nucleus

## Data Availability

Not applicable.

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
