# Peer review of "Spontaneous Occurrence of Various Types of Hepatocellular Adenoma in the Livers of Metabolic Syndrome-Associated Steatohepatitis Model TSOD Mice"

_ijms, 2022, doi:10.3390/ijms231911923_

Round 1

Reviewer 1 Report

In the work entitled: “Spontaneous occurrence of various types of hepatocellular adenoma in the livers of metabolic syndrome-associated steatohepatitis model TSOD mice,” the authors validate a mouse model of human hepatocellular adenoma.

They use, for their validation, several biochemical markers such as HNF1A, immune activation via IL-6/JAK/STAT3, and b-catenin to make comparisons between the behavior of the human tumor and the tumor of the DSOT mice, a model of nonalcoholic steatohepatitis, from a spontaneous syndrome in mice previously reported by the authors.

Their results describe the frequency of tumor formation and their anatomy in the mouse model and present a histochemical analysis of several tumors, which supports the validation of their hepatocellular adenoma model. The work is well written, the results are precise, and the interpretation is accessible to the reader.

It would be desirable to have the histological analysis parallel between mouse and human tissue so that the reader could make their interpretation; however, the data is clear enough that this point is not a conditioning factor for publication. On the other hand, the article's novelty decreases because it is a second stage in which they are trying to validate the model already described.

Author Response

To Reviewer #1

Thank you for your valuable comments. We appreciate the time spent by you and believe the revised manuscript is improved. Below, we have addressed the reviewers’ comments.

It would be desirable to have the histological analysis parallel between mouse and human tissue so that the reader could make their interpretation; however, the data is clear enough that this point is not a conditioning factor for publication. On the other hand, the article's novelty decreases because it is a second stage in which they are trying to validate the model already described.

[Response]

Thank you for your comments. In this article, we used mice tissue staining comparisons to analyze various histological types. As you pointed out clearly demonstrating the similarities and differences with human tissue is very important to highlight the value of expanding the use of this mouse in the future. We are currently using additional TSOD mice that will be re-examined at 6 months (24W), 8 months (32W), 10 months (40W) and 12 months (48W) (10 mice each group). Finally, we are planning additional studies (including comprehensive genetic analysis) to elucidate similarities with humans.

Reviewer 2 Report

In this article entitled 'Spontaneous occurrence of various types of hepatocellular adenoma in the livers of metabolic syndrome-associated steatohepatitis model TSOD mice', the authors examined the characteristics of GS-negative tumors that spontaneously develop in TSOD mice against the background of metabolic syndrome. Here I list  my comments.

1. Please check if all background information has been supported by citations. For example, the references were missing for contents in sections 1.1-1.4.

2. The informtaion of data analysis was missing.

3. The data description of Figure 2 was missing.

4.  A few of scale bars in histological figures were so blurry.

5. The conclusion was not clearly presented.

6. Please unify the word use of HNF1α.

7. Please define TSOD when it's first used. 

8. What is the control for TSOD mice?

9. What is the general hypothesis of this study? This was not clearly stated in either introduction or answered in discussion.

Author Response

Reply for the Reviewer 2

Thank you for your valuable comments. We appreciate the time spent by you and believe the revised manuscript is improved. Below, we have addressed the reviewers’ comments.

  1. Please check if all background information has been supported by citations. For example, the references were missing for contents in sections 1.1-1.4.

Thanks for your point. We have added references#18,19,21,22 in the reference list.

  1. The informtaion of data analysis was missing.

Thanks for your point. The present study examined the pathological character of six GS-negative tumors found only in TSOD mice and did not perform any data analysis that would require statistical processing. We are currently working on comparing the frequency of GS-negative tumors by subclass with the frequency of GS-positive tumors by observing TSOD mice over time. Statistical analysis will be performed when we compile the results of these studies.

  1. The data description of Figure 2 was missing.

Thanks for your point. Please see ”3.3.1. HNF1α-inactivated HCA tumor mimics” paragraph (L162-169) in the Result.

  1. A few of scale bars in histological figures were so blurry.

Thanks for your point. We have changed the dashed yellow line describing the size of the tumor to black. Also, the direction of the HE staining image rotated to unify the immunostaining images in Figure 2G and Figure 3G.

  1. The conclusion was not clearly presented.

Thank you for your point. We summarized the points and added conclusions at the end of the discussion section.

  1. Please unify the word use of HNF1α.

We modified the format of HNF1α and confirmed that all formats are consistent.

  1. Please define TSOD when it's first used.

And

  1. What is the control for TSOD mice?

TSOD mice, have been added with full spelling and described the creation method. TSNO mice are control mice, and their creation method has been added. Throughout the experiment, none of the liver tumors developed in the TSNO mice group.

  1. What is the general hypothesis of this study? This was not clearly stated in either introduction or answered in discussion.

 In the final part of the Introduction, we have stated the general hypothesis of this study and the significance of this study.

Round 2

Reviewer 2 Report

My comments have been addressed.